# End-to-end Symbolic Regression with Transformers

**Pierre-Alexandre Kamienny**[*1,2], **Stéphane d'Ascoli**[*1,3]
**Guillaume Lample**[1], **François Charton**[1]
[1]Meta AI
[2]ISIR MLIA, Sorbonne Université
[3]Department of Physics, Ecole Normale Supérieure
`pakamienny@meta.com`

## Abstract

Symbolic regression, the task of predicting the mathematical expression of a function from the observation of its values, is a difficult task which usually involves a two-step procedure: predicting the "skeleton" of the expression up to the choice of numerical constants, then fitting the constants by optimizing a non-convex loss function. The dominant approach is genetic programming, which evolves candidates by iterating this subroutine a large number of times. Neural networks have recently been tasked to predict the correct skeleton in a single try, but remain much less powerful.

In this paper, we challenge this two-step procedure, and task a Transformer to directly predict the full mathematical expression, constants included. One can subsequently refine the predicted constants by feeding them to the non-convex optimizer as an informed initialization. We present ablations to show that this end-to-end approach yields better results, sometimes even without the refinement step. We evaluate our model on problems from the SRBench benchmark and show that our model approaches the performance of state-of-the-art genetic programming with several orders of magnitude faster inference.

## Introduction

Inferring mathematical laws from experimental data is a central problem in natural science; having observed a variable $y$ at $n$ points $\{x_i\}_{i \in \mathbb{N}_n}$, it implies finding a function $f$ such that $y_i \approx f(x_i)$ for all $i \in \mathbb{N}_n$. Two types of approaches exist to solve this problem. In *parametric statistics* (PS), the function $f$ is defined by a small number of parameters that can directly be estimated from the data. On the other hand, *machine learning* (ML) techniques such as decision trees and neural networks select $f$ from large families of non-linear functions by minimizing a loss over the data. The latter relax the assumptions about the underlying law, but their solutions are more difficult to interpret, and tend to overfit small experimental data sets, yielding poor extrapolation performance.

Symbolic regression (SR) stands as a middle ground between PS and ML approaches: $f$ is selected from a large family of functions, but is required to be defined by an interpretable analytical expression. It has already proved extremely useful in a variety of tasks such as inferring physical laws [1, 2].

SR is usually performed in two steps. First, predicting a "skeleton", a parametric function using a pre-defined list of operators – typically, the basic operations $(+, \times, \div)$ and functions $(\mathrm{sqrt}, \exp, \sin,$ etc.). It determines the general shape of the law up to a choice of constants, e.g. $f(x) = \cos(ax + b)$. Then, the constants in the skeleton $(a, b)$ are estimated using optimization techniques, typically the Broyden–Fletcher–Goldfarb–Shanno algorithm (BFGS).

---

[*]equal contribution

36th Conference on Neural Information Processing Systems (NeurIPS 2022).

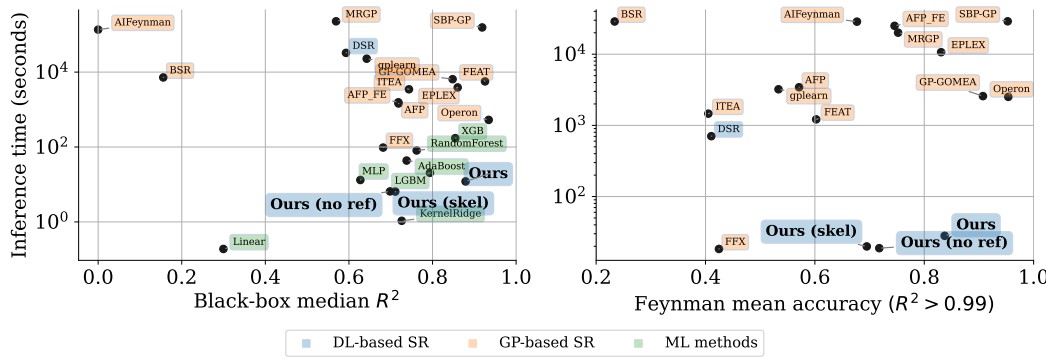

Figure 1: **Our model outperforms previous DL-based methods and offers at least an order of magnitude inference speedup compared to SOTA GP-based methods.** Pareto plot comparing the average test performance and inference time of our models with baselines provided by the SRbench benchmark [7], both on Feynman SR problems [1] and black-box regression problems. We use colors to distinguish three families of models: **deep-learning based SR**, **genetic programming-based SR** and **classic machine learning methods** (which do not provide symbolic solutions). A similar Pareto plot against formula complexity is provided in Fig. 11.

The leading algorithms for SR rely on genetic programming (GP). At each generation, a population of candidates is predicted, and the fittest ones are selected based on the data, and mutated to build the next generation. The algorithm iterates this procedure until a satisfactory accuracy level is achieved.

While GP algorithms achieve good prediction accuracy, they are notably slow (see the Pareto plot of Fig. 1). Indeed, the manually predefined function space to search is generally vast, and each generation involves a costly call to the BFGS routine. Also, GP does not leverage past experience: every new problem is learned from scratch. Inference time, i.e. time required to output a satisfactory expression, for most GP algorithms is both long and unbounded (the longer, the better the results), therefore this property has been neglected by the SR community, however fast inference can be useful to improve search-based SR algorithms, e.g. by reducing search space or providing initial guesses, as well as to tackle practical applications with time constraints, e.g. control or reinforcement learning [3, 4].

Supervised training neural networks built for language modelling on large datasets of synthetic examples has recently been proposed for SR [5, 6]. These references follow the two-step procedure (predicting the skeleton then fitting the constants) inherited from GP. Once the model is trained, at inference, the skeleton is predicted via a simple forward pass, and a single call to BFGS is needed, thus resulting in a significant speed-up compared to GP. However, these methods are not as accurate as state-of-the-art GP, and have so far been limited to low-dimensional functions ($D \leq 3$). We argue that two reasons underlie their shortcomings.

First, skeleton prediction is an ill-posed problem that does not provide sufficient supervision: different instances of the same skeleton can have very different shapes, and instances of very different skeletons can be very close. Second, the loss function minimized by BFGS can be highly non-nonconvex: even when the skeleton is perfectly predicted, the correct constants are not guaranteed to be found. For these reasons, we believe, and will show, that doing away with skeleton estimation as a intermediary step can greatly facilitate the task of SR for language models.

**Contributions** In this paper, we train Transformers over synthetic datasets to perform **end-to-end (E2E)** symbolic regression: solutions are predicted directly, without resorting to skeletons. To this effect, we leverage a hybrid **symbolic-numeric vocabulary**, that uses both symbolic tokens for the operators and variables and numeric tokens for the constants. One can then perform a **refinement** of the predicted constants by feeding them as informed guess to BFGS, mitigating non-linear optimization issues. Finally, we introduce **generation and inference techniques** that allow our models to scale to larger problems: up to 10 input features against 3 in concurrent works.

Evaluated over the SRBench benchmark [7], our model significantly narrows the accuracy gap with state-of-the-art GP techniques, while providing several orders of magnitude of inference time speedup (see Fig. 1). We also demonstrate strong robustness to noise and extrapolation capabilities.

**Related work** SR is a challenging task that traces back from a few decades ago, with a large number of open-source and commercial softwares, and has already been used to accelerate scientific discoveries [8, 9, 10]. Most popular frameworks for symbolic regression use GP [11, 12, 13, 14, 15, 16, 17, 18, 19] (see [7] for a recent review), but SR has also seen growing interest from the Deep Learning (DL) community, motivated by the fact that neural networks are good at identifying qualitative patterns.

Neural networks have been combined with GP algorithms, e.g. to simplify the original dataset [1], or to propose a good starting distribution over mathematical expressions[20]. [21, 22] propose modifications to feed-forward networks to include interpretable components, i.e. replacing usual activation functions by operators such as $\cos, \sin$, however these are hard to optimize and prone to numerical issues.

Language models, and especially Transformers [23], have been trained over synthetic datasets to solve various mathematical problems: integration [24], dynamical systems [25], linear algebra [26], formal logic [27] and theorem proving [28]. A few papers apply these techniques to symbolic regression: the aforementioned references [6, 5] train Transformers to predict function skeletons, while [29] infers one-dimensional recurrence relations in sequences of numbers. [30] trains fully-connected networks to predict simple formulas from tabular data.

The recently introduced SRBench [7] provides a benchmark for rigorous evaluation of SR methods, in addition to 14 SR methods and 7 ML baselines which we will compare to in this work.

# 1 Data generation

Our approach consists in training language models on vast synthetic datasets. Each training example is a pair: a set of $N$ points $(x, y) \in \mathbb{R}^D \times \mathbb{R}$ as the input, and a function $f$ such that $y = f(x)$ as the target[2] Examples are generated by first sampling a random function $f$, then a set of $N$ input values $(x_i)_{i \in \mathbb{N}_N}$ in $\mathbb{R}^D$, and computing $y_i = f(x_i)$.

## 1.1 Generating functions

To sample functions $f$, we follow the seminal approach of Lample and Charton [24], and generate random trees with mathematical operators as internal nodes and variables or constants as leaves. The procedure is detailed below (see Table 3 in the Appendix for the values of parameters):

1. Sample the desired **input dimension** $D$ of the function $f$ from $\mathcal{U}\{1, D_{\max}\}$.
2. Sample the number of **binary operators** $b$ from $\mathcal{U}\{D - 1, D + b_{\max}\}$ then sample $b$ operators from $\mathcal{U}\{+, -, \times\}$[3].
3. Build a **binary tree** with those $b$ nodes, using the sampling procedure of [24].
4. For each **leaf** in the tree, sample one of the variables $x_d$, $d \in \mathbb{N}_D$.
5. Sample the number of **unary operators** $u$ from $\mathcal{U}\{0, u_{\max}\}$ then sample $u$ operators from the list $O_u$ in Table 3, and insert them at random positions in the tree.
6. For each variable $x_d$ and unary operator $u$, apply a random **affine transformation**, i.e. replace $x_d$ by $ax_d + b$, and $u$ by $au + b$, with $(a, b)$ sampled from $\mathcal{D}_{\text{aff}}$.

Note that since we require independent control on the number of unary operators (which is independent of $D$) and binary operators (which depends on $D$), we cannot directly sample a unary-binary tree as in [24]. Note also that the first $D$ variables are sampled in ascending order to obtain the desired input dimension, which means functions with missing variables such as $x_1 + x_3$ are never encountered; this is not an issue as our model can always set the prefactor of $x_2$ to zero. As discussed quantitatively

---

[2]We only consider functions from $\mathbb{R}^D$ into $\mathbb{R}$; the general case $f : \mathbb{R}^D \rightarrow \mathbb{R}^P$ can be handled as $P$ independent subproblems.

[3]Note that although the division operation is technically a binary operator, it appears much less frequently than additions and multiplications in typical expressions [31], hence we replace it by the unary operator inv: $x \rightarrow 1/x$.

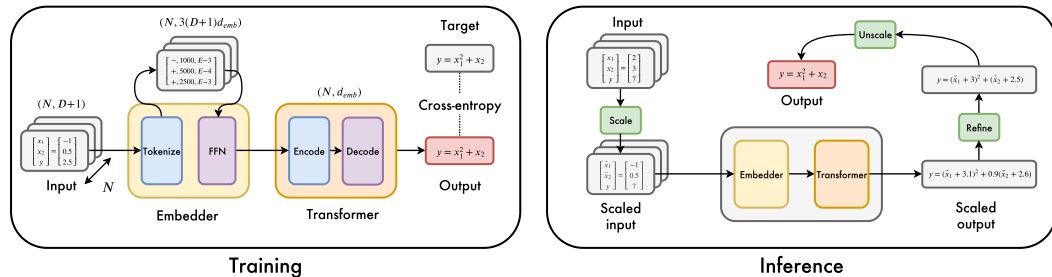

Figure 2: **Sketch of our model.** During training, the inputs are all whitened. At inference, we whiten them as a pre-processing step; the predicted function must then be unscaled to account for the whitening.

in App. C, the number of possible skeletons as well as the random sampling of numerical constants guarantees that our model almost never sees the same function twice, and cannot simply perform memorization. See App. B for examples of the skeleton of generated expressions.

## 1.2 Generating inputs

For each function $f : \mathbb{R}^D \to \mathbb{R}$, we sample $N \in \mathcal{U}\{10D, N_{\max}\}$ **input values** $x_i \in \mathbb{R}^D$ from the distribution $\mathcal{D}_x$ described below, and compute the corresponding **output values** $y_i = f(x_i)$. If any $x_i$ is outside the domain of definition of $f$ or if any $y_i$ is larger $10^{100}$, the process is aborted, and we start again by generating a new function. Note that rejecting and resampling out-of-domain values of $x_i$, the obvious and cheaper alternative, would provide the model with additional information about $f$, by allowing it to learn its domain of definition.

To maximize the diversity of input distributions seen during training, we sample our inputs from a mixture of distributions (uniform or gaussian), centered around $k$ random centroids[4], see App. A for some illustrations at $D = 2$. Input samples are generated as follows:

1. Sample a **number of clusters** $k \sim \mathcal{U}\{1, k_{max}\}$ and $k$ **weights** $w_i \sim \mathcal{U}(0, 1)$, which are then normalized so that $\sum_i w_i = 1$.
2. For each cluster $i \in \mathbb{N}_k$, sample a **centroid** $\mu_i \sim \mathcal{N}(0, 1)^D$, a vector of **variances** $\sigma_i \sim \mathcal{U}(0, 1)^D$ and a **distribution shape** (gaussian or uniform) $\mathcal{D}_i \in \{\mathcal{N}, \mathcal{U}\}$.
3. For each cluster $i \in \mathbb{N}_k$, sample $\lfloor w_i N \rfloor$ **input points** from $\mathcal{D}_i(\mu_i, \sigma_i)$ then apply a **random rotation** sampled from the Haar distribution.
4. Finally, **concatenate** all the points obtained and **whiten** them by substracting the mean and dividing by the standard deviation along each dimension.

## 1.3 Tokenization

Following [26], we represent numbers in base 10 floating-point notation, round them to four significant digits, and encode them as sequences of 3 tokens: their sign, mantissa (between 0 and 9999), and exponent (from `E-100` to `E100`).

To represent mathematical functions as sequences, we enumerate the trees in prefix order, i.e. direct Polish notation, as in [24]: operators and variables and integers are represented as single autonomous tokens, and constants are encoded as explained above.

For example, the expression $f(x) = \cos(2.4242x)$ is encoded as `[cos,mul,+,2424,E-3,x]`. Note that the vocabulary of the decoder contains a mix of symbolic tokens (operators and variables) and numeric tokens, whereas that of the encoder contains only numeric tokens[5].

---

[4]For $k \to \infty$, such a mixture could in principe approximate any input distribution.

[5]The embeddings of numeric tokens are *not* shared between the encoder and decoder.

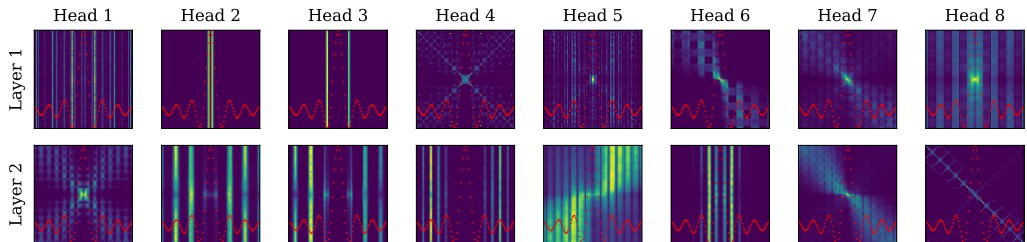

Figure 3: **Attention heads reveal intricate mathematical analysis.** We considered the expression $f(x) = \sin(x)/x$, with $N = 100$ input points sampled between $-20$ and $20$ (red dots; the y-axis is arbitrary). We plotted the attention maps of a few heads of the encoder, which are $N \times N$ matrices where the element $(i, j)$ represents the attention between point $i$ and point $j$. Notice that heads 2, 3 and 4 of the second layer analyze the periodicity of the function in a Fourier-like manner.

## 2 Methods

Below we describe our approach for end-to-end symbolic regression; please refer to Fig. 2 for an illustration.

### 2.1 Model

**Embedder**   Our model is provided $N$ input points $(x, y) \in \mathbb{R}^{D+1}$, each of which is represented as $3(D + 1)$ tokens of dimension $d_{\text{emb}}$. As $D$ and $N$ become large, this results in long input sequences (e.g. 6600 tokens for $D = 10$ and $N = 200$), which challenge the quadratic complexity of Transformers. To mitigate this, we introduce an embedder to map each input point to a single embedding.

The embedder pads the empty input dimensions to $D_{\text{max}}$, then feeds the $3(D_{\text{max}} + 1)d_{\text{emb}}$-dimensional vector into a 2-layer fully-connected feedforward network (FFN) with ReLU activations, which projects down to dimension $d_{\text{emb}}$[6] The resulting $N$ embeddings of dimension $d_{\text{emb}}$ are then fed to the Transformer.

**Transformer**   We use a sequence to sequence Transformer architecture [23] with 16 attention heads and an embedding dimension of 512, containing a total of 86M parameters. Like [26], we observe that the best architecture for this problem is asymmetric, with a deeper decoder: we use 4 layers in the encoder and 16 in the decoder. A notable property of this task is the permutation invariance of the $N$ input points. To account for this invariance, we remove positional embeddings from the encoder.

As shown in Fig. 3 and detailed in App. D, the encoder captures the most distinctive features of the functions considered, such as critical points and periodicity, and blends a mix of short-ranged heads focusing on local details with long-ranged heads which capture the global shape of the function.

**Training**   We optimize a cross-entropy loss with the Adam optimizer, warming up the learning rate from $10^{-7}$ to $2.10^{-4}$ over the first 10,000 steps, then decaying it as the inverse square root of the number of steps, following [23]. We hold out a validation set of $10^4$ examples from the same generator, and train our models until the accuracy on the validation set saturates (around 50 epochs of 3M examples).

Input sequence lengths vary significantly with the number of points $N$; to avoid wasteful padding, we batch together examples of similar lengths, ensuring that a full batch contains a minimum of 10,000 tokens. On 32 GPU with 32GB memory each, one epoch is processed in about half an hour.

### 2.2 Inference tricks

In this section, we describe three tricks to improve the performance of our model at inference.

---

[6]We explored various architectures for the embedder, but did not obtain any improvement; this does not appear to be a critical part of the model.

Table 1: The importance of an end-to-end model with refinement.

| Model | Function $f(x, y)$ |
|-------|--------------------|
| Target | $\sin(10x) \exp(0.1y)$ |
| Skeleton + BFGS | $-\sin(1.7x)(0.059y + 0.19)$ |
| E2E no BFGS | $\sin(9.9x) \exp(0.1y)$ |
| E2E + BFGS random init | $-\sin(0.095x) \exp(0.27y)$ |
| E2E + BFGS model init | $\sin(10x) \exp(0.1y)$ |

The skeleton approach recovers an incorrect skeleton. The E2E approach predicts the right skeleton. Refinement worsens original prediction when randomly initialized, and yields the correct result when initialized with predicted constants.

**Refinement**  Previous language models for SR, such as [6], follow a *skeleton* approach: they first predict equation skeletons, then fit the constants with a non-linear optimisation solver such as BFGS. In this paper, we follow an *end-to-end* (E2E) approach: predicting simultaneously the function and the values of the constants. However, we improve our results by adding a *refinement* step: fine-tuning the constants a posteriori with BFGS, initialized with our model predictions[7].

This results in a large improvement over the skeleton approach, as we show by training a Transformer to predict skeletons in the same experimental setting. The improvement comes from two reasons: first, prediction of the full formula provides better supervision, and helps the model predict the skeleton; second, the BFGS routine strongly benefits from the informed initial guess, which helps the model predict the constants. This is illustrated qualitatively in Table 1, and quantitatively in Table 2.

**Scaling**  As described in Section 1.2, all input points presented to the model during training are whitened: their distribution is centered around the origin and has unit variance. To allow accurate prediction for input points with a different mean and variance, we introduce a scaling procedure during inference. Let $f$ the function to be inferred, $x$ be the input points, and $\mu = \text{mean}(x), \sigma = \text{std}(x)$. As illustrated in Fig. 2 we pre-process the input data by replacing $x$ by $\tilde{x} = \frac{x-\mu}{\sigma}$. The model then predicts $\hat{f}(\tilde{x}) = \hat{f}(\sigma x + \mu)$, and we can recover an approximation of $f$ by unscaling the variables in $\hat{f}$.

This gives our model the desirable property to be insensitive to the scale of the input points: DL-based approaches to SR are known to fail when the inputs are outside the range of values seen during training [29, 26]. Note that here, the scale of the inputs translates to the scale of the constants in the function $f$; although these coefficients are sampled in $\mathcal{D}_{\text{aff}}$ during training, coefficients outside $\mathcal{D}_{\text{aff}}$ can be expressed by multiplication of constants in $\mathcal{D}_{\text{aff}}$.

**Bagging and decoding**  Since our model was trained on $N \leq 200$ input points, it does not perform satisfactorily at inference when presented with more than 200 input points. To take advantage of large datasets while accommodating memory constraints, we perform *bagging*: whenever $N$ is larger than 200 at inference, we randomly split the dataset into $B$ bags of 200 input points[8].

For each bag, we apply a forward pass and generate $C$ function candidates via random sampling or beam search using the next token distribution. As shown in App. F (Fig. 16), the more commonly used beam search [34] strategy leads to much less good results than sampling due to the lack of diversity induced by constant prediction (typical beams will look like $\sin(x), \sin(1.1x), \sin(0.9x), \ldots$). This provides us with a set of $BC$ candidate solutions.

**Inference time**  Our model inference speed has two sources: the forward passes described above on one hand (which can be parallelized up to memory limits of the GPU), and the refinements of candidate functions on the other (which are CPU-based and could also be parallelized, although we did not consider this option here).

---

[7]To avoid BFGS having to approximate gradients via finite differences, we provide the analytical expression of the gradient using *sympytorch* [32] and *functorch* [33].

[8]Smarter splits, e.g. diversity-preserving, could be envisioned, but were not considered here.

Table 2: Our approach outperforms the skeleton approach.

| Model | $R^2$ | $\text{Acc}_{0.1}$ | $\text{Acc}_{0.01}$ | $\text{Acc}_{0.001}$ |
|---|---|---|---|---|
| Skeleton + BFGS | 0.43 | 0.40 | 0.27 | 0.17 |
| E2E no BFGS | 0.62 | 0.51 | 0.27 | 0.09 |
| E2E + BFGS random init | 0.44 | 0.44 | 0.30 | 0.19 |
| E2E + BFGS model init | **0.68** | **0.61** | **0.44** | **0.29** |

Metrics are computed over the $10,000$ examples of the evaluation set.

Since $BC$ can become large, we rank candidate functions (according to their error on *all* input points), get rid of redundant skeleton functions and keep the best $K$ candidates for the refinement step[9]. To speed up the refinement, we use a subset of at most $1024$ input points for the optimization. The parameters $B$, $C$ and $K$ can be used as cursors in the speed-accuracy tradeoff: in the experiments presented in Fig. 1, we selected $B = 100$, $C = 10$, $K = 10$.

## 3 Results

In this section, we present the results of our model. We begin by studying in-domain accuracy, then present results on out-of-domain datasets.

### 3.1 In-domain performance

We report the in-domain performance of our models by evaluating them on a fixed validation set of 100,000 examples, generated as per Section 1. Validation functions are uniformly spread out over three difficulty factors: number of unary operators, binary operators, and input dimension. For each function, we evaluate the performance of the model when presented $N = [50, 100, 150, 200]$ input points $(x, y)$, and prediction accuracy is evaluated on $N_{\text{test}} = 200$ points sampled from a fresh instance of the multimodal distribution described in Section 1.2.

We assess the performance of our model using two popular metrics: $R^2$-score [7] and accuracy to tolerance $\tau$ [6, 29]:

$$R^2 = 1 - \frac{\sum_i^{N_{\text{test}}} (y_i - \hat{y}_i)^2}{\sum_i^{N_{\text{test}}} (y_i - \bar{y})^2}, \qquad \text{Acc}_\tau = \mathbb{1} \left( \max_{1 \leq i \leq N_{\text{test}}} \left| \frac{\hat{y}_i - y_i}{y_i} \right| \leq \tau \right), \qquad (1)$$

where $\mathbb{1}$ is the indicator function.

$R^2$ is classically used in statistics, but it is unbounded, hence a single bad prediction can cause the average $R^2$ over a set of examples to be extremely bad. To circumvent this, we set $R^2 = 0$ upon pathological examples as in [7](such examples occur in less that 1% of cases)[10]. The accuracy metric provides a better idea of the precision of the predicted expression as it depends on a desired tolerance threshold. However, due to the presence of the max operator, it is sensitive to outliers, and hence to the number of points considered at test time (more points entails a higher risk of outlier). To circumvent this, we discard the 5% worst predictions, following [6].

**End-to-end outperforms skeleton** In Table 2, we report the average in-domain results of our models. Without refinement, our E2E model outperforms the skeleton model trained under the same protocol in terms of low precision prediction ($R^2$ and $\text{Acc}_{0.1}$ metrics), but small errors in the prediction of the constants lead to lower performance at high precision ($\text{Acc}_{0.001}$ metric). The refinement procedure alleviates this issue significantly, inducing a three-fold increase in $\text{Acc}_{0.001}$ while also boosting other metrics. Initializing BFGS with the constants estimated in the E2E phase plays a crucial role: with random initialization, the BFGS step actually *degrades* E2E performance. However, refinement with random initialization still achieves better results than the skeleton model: this suggests that the E2E model predicts skeletons better that the skeleton model.

---

[9]Though these candidates are the best functions without refinement, there are no guarantees that these would be the best after refinement, especially as optimization is particularly prone to spurious local optimas.

[10]Note that predicting the constant function $f = \bar{y}$ naturally yields an $R^2$ score of 0.

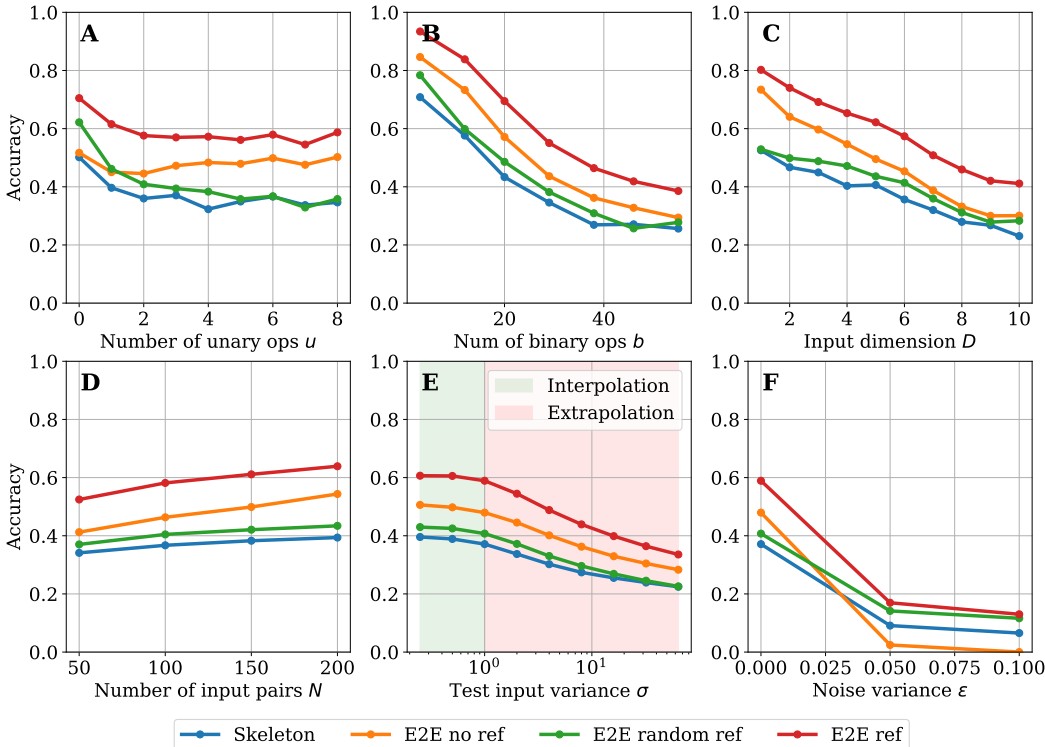

Figure 4: **Ablation over the function difficulty (top row) and input difficulty (bottom row).** We plot the accuracy at $\tau = 0.1$ (Eq. 1), see App. E for the $R^2$ score. We distinguish four models: **skeleton**, **E2E without refinement**, **E2E with refinement from random guess** and **E2E with refinement**. **A:** number of unary operators. **B:** number of binary operators. **C:** input dimension. **D:** Low-resource performance, evaluated by varying the number of input points. **E:** Extrapolation performance, evaluated by varying the variance of the inputs. **F:** Robustness to noise, evaluated by varying the multiplicative noise added to the labels.

**Ablation** Fig. 4A,B,C presents an ablation over three indicators of formula difficulty (from left to right): number of unary operators, number of binary operators and input dimension. In all cases, increasing the factor of difficulty degrades performance, as one could expect. This may give the impression that our model does not scale well with the input dimension, but we show that our model scales in fact very well on out-of-domain datasets compared to concurrent methods (see Fig. 15 of the Appendix). We include a qualitative ablation on the improvement caused by the use of mixture of distributions in App. E.

Fig. 4D shows how performance depends on the number of input points fed to the model, $N$. In all cases, performance increases, but much more signicantly for the E2E models than for the skeleton model, demonstrating the importance of having a lot of data to accurately predict the constants in the expression.

**Extrapolation and robustness** In Fig. 4E, we examine the ability of our models to interpolate/extrapolate by varying the scale of the test points: instead of normalizing the test points to unit variance, we normalize them to a scale $\sigma$. As expected, performance degrades as we increase $\sigma$, however the extrapolation performance remains decent even very far away from the inputs ($\sigma = 32$).

Finally, in Fig. 4F, we examine the effect of corrupting the targets $y$ with a multiplicative noise of variance $\sigma$: $y \rightarrow y(1 + \xi), \xi \sim \mathcal{N}(0, \varepsilon)$. The results reveal something interesting: without refinement, the E2E model is not robust to noise, and actually performs worse than the skeleton model at high noise. This shows how sensitive the Transformer is to the inputs when predicting constants. Refinement improves robustness significantly, but the initialization of constants to estimated values has less impact, since the prediction of constants is corrupted by the noise.

## 3.2  Out-of-domain generalization

We evaluate our method on the recently released benchmark SRBench[7]. Its repository contains a set of $252$ regression datasets from the Penn Machine Learning Benchmark (PMLB)[35] in addition to $14$ open-source SR and ML baselines. The datasets consist in "ground-truth" problems where the true underlying function is known, as well as "black-box" problems which are more general regression datasets without an underlying ground truth. We filter out problems from SRBench to only keep regression problems with $D \leq 10$ with continuous features; this results in $190$ regression datasets, splitted into 57 black-box problems (combination of real-world and noisy, synthetic datasets), $119$ SR datasets from the Feynman [1] and $14$ SR datasets from the ODE-Strogatz [36] databases. Each dataset is split into $75\%$ training data and $25\%$ test data, on which performance is evaluated.

The overall performance of our models is illustrated in the Pareto plot of Fig. 1, where we see that on both types of problems, our model achieves performance close to state-of-the-art GP models such as Operon with a fraction of the inference time[11]. Impressively, our model outperforms all classic ML methods (e.g. XGBoost and Random Forests) on real-world problems with a lower inference time, and while outputting an interpretable formula.

We provide more detailed results on Feynman problems in Fig. 5, where we additionally plot the formula complexity, i.e. the number of nodes in the mathematical tree (see App. F for similar results on black-box and Strogatz problems). Varying the noise applied to the targets noise, we see that our model displays similar robustness to state-of-the-art GP models. We additionally include ablation on the use of scaling during inference in App. E.

While the average accuracy or our model is only ranked fourth, it outputs formulas with lower complexity than the top 2 models (Operon and SBP-GP), which is an important criteria for SR problems: see App. 11 for complexity-accuracy Pareto plots. To the best of our knowledge, our model is the first non-GP approach to achieve such competitive results for SR.

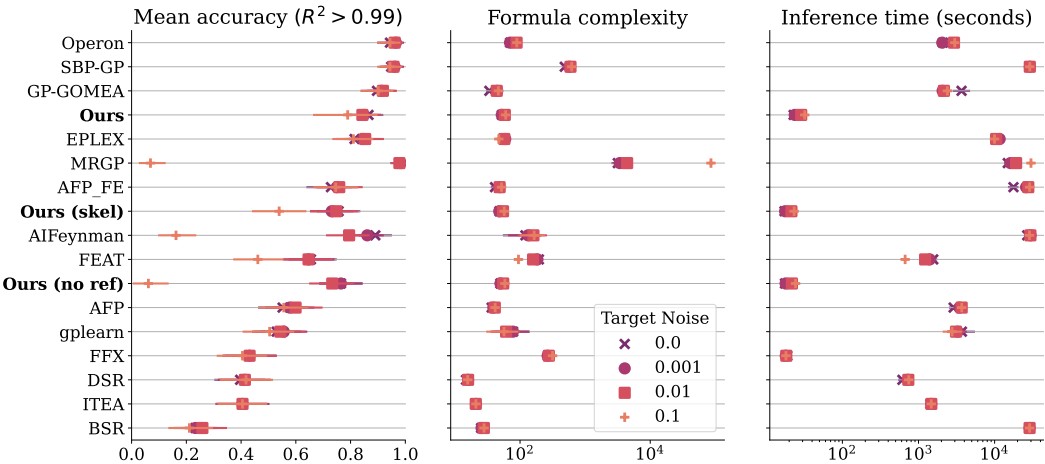

Figure 5: **Our model presents strong accuracy-speed-complexity tradeoffs, even in presence of noise.** Results are averaged over all 119 Feynman problems, for 10 random seeds and three target noises each as shown in the legend. The accuracy is computed as the fraction of problems for which the $R^2$ score on test examples is above 0.99. Models are ranked according to the accuracy averaged over all target noise.

## Conclusion

In this work, we introduced a competitive deep learning model for SR by using a novel numeric-symbolic approach. Through rigorous ablations, we showed that predicting the constants in an expression not only improves performance compared to predicting a skeleton, but can also serve as an informed initial condition for a solver to refine the value of the constants.

---

[11]Inference uses a single GPU for the forward pass of the Transformer.

Our model outperforms previous deep learning approaches by a margin on SR benchmarks, and scales to larger dimensions. Yet, the dimensions considered here remain moderate ($D < 10$): adapting to the truly high-dimensional setup is an interesting future direction, and will likely require qualitative changes in the data generation protocol. While our model narrows the gap between GP and DL based SR, closing the gap also remains a challenge for future work.

This work opens up a whole new range of applications for SR in fields which require real-time inference. We hope that the methods presented here may also serve as a toolbox for many future applications of Transformers for symbolic tasks.

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
