# A  Details on the training data

In Tab. 3 we provide the detailed set of parameters used in our data generator. The probabilities of the unary operators were selected to match the natural frequencies appearing in the Feynman dataset.

In Fig. 6, we show the statistics of the data generation.The number of expressions diminishes with the input dimension and number of unary operators because of the higher likelihood of generating out-of-domain inputs. One could easily make the distribution uniform by enforcing to retry as long as a valid example is not found, however we find empirically that having more easy examples than hard ones eases learning and provides better out-of-domain generalization, which is our ultimate goal.

In Fig. 7, we show some examples of the input distributions generated by our multimodal approach. Notice the diversity of shapes obtained by this procedure.

Table 3: **Parameters of our generator.**

| Parameter | Description | Value |
|---|---|---|
| $D_{\max}$ | Max input dim | 10 |
| $\mathcal{D}_{\mathrm{aff}}$ | Distrib of $(a,b)$ | sign $\sim \mathcal{U}\{-1, 1\}$, mantissa $\sim \mathcal{U}(0, 1)$, exponent $\sim \mathcal{U}(-2, 2)$ |
| $b_{\max}$ | Max binary ops | $5 + D$ |
| $O_b$ | Binary operators | add:1, sub:1, mul:1 |
| $u_{\max}$ | Max unary ops | 5 |
| $O_u$ | Unary operators | inv:5, abs:1, sqr:3, sqrt:3, sin:1, cos:1, tan:0.2, atan:0.2, log:0.2, exp:1 |
| $N_{\min}$ | Min number of points | $10D$ |
| $N_{\max}$ | Max number of points | 200 |
| $k_{max}$ | Max num clusters | 10 |

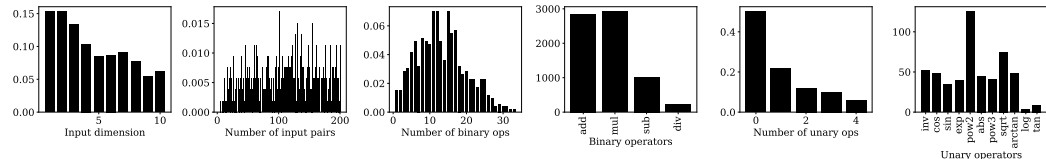

Figure 6: **Statistics of the synthetic data.** We calculated the latter on $10,000$ generated examples.

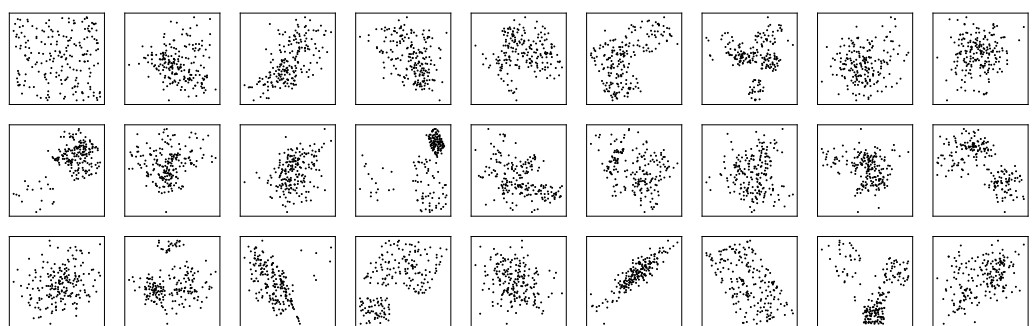

Figure 7: **Diversity of the input distributions generated by the multimodal approach.** Here we show distributions obtained for $D = 2$.

## B  Examples of expressions generated

Below, we give some typical examples of the expressions generated by our random generator for dimensions between 1 and 3 (Tab. 4), as well as some examples of equations from the Feynman dataset (Tab. 5).

$$Cx_0 - \frac{Cx_0}{Cx_0+C} - C\left(Cx_0\left(Cx_0+C\right)^3 + C\right)^3 + C$$

$$Cx_0 + C\left|Cx_0 + C\right| + C$$

$$Cx_0x_1\left(Cx_0 + C\log\left(Cx_0+C\right)\right) + C$$

$$C + x_0\left(-Cx_1 + Cx_2\right)^2$$

$$C + x_0x_2\left(Cx_0 + Cx_1 - C\sin\left(Cx_0 - C\sin\left(Cx_1x_2 + C\right)\right)\right)$$

$$Cx_0 + Cx_2 + C + \frac{C}{Cx_1\sqrt{Cx_0+C} + Ce^{Cx_0 - \frac{C}{Cx_2+C}}}$$

$$Cx_0 + Cx_1 + C$$

$$C\left(Cx_0 + C\right)^3 + C - \frac{C}{\frac{Cx_1}{Cx_2+C}+C}$$

$$C\sin\left(Cx_0 + C\right) + C + \frac{C}{Cx_1+C}$$

$$-Cx_1\left(\frac{Cx_2}{Cx_0+C} + C\right)^2 + Cx_2 + C\sin\left(Cx_1+C\right) + C\tan\left(Cx_0+C\right) + C$$

$$C + x_0\left(Cx_0 - Cx_1 + C\left|Cx_1 + C\right|\right)$$

$$C + x_2\left(Cx_0 + Cx_1\right)\left(Cx_0 + \frac{C}{Cx_0+C}\right)$$

$$C - x_2\sin\left(Cx_0 + Cx_1 - \frac{Cx_2}{Cx_2+C}\right)$$

$$-Cx_0 + C\left(\frac{Cx_0}{Cx_0+C} + C\right)^2 + C$$

$$Cx_0\left(Cx_0x_1x_2 + \frac{C}{Cx_2+C}\right) + Cx_1 + C\left(Cx_1+C\right)^2 + C$$

Table 4: A few examples of equations from our random generator.

| | | |
|---|---|---|
| $\dfrac{\sqrt{2}e^{-\frac{\theta^2}{2}}}{2\sqrt{\pi}}$ | $\dfrac{\sqrt{2}e^{-\frac{\theta^2}{2\sigma^2}}}{2\sqrt{\pi}\sigma}$ | $\dfrac{\sqrt{2}e^{-\frac{(\theta-\theta_1)^2}{2\sigma^2}}}{2\sqrt{\pi}\sigma}$ |
| $\sqrt{\left(-x_1+x_2\right)^2 + \left(-y_1+y_2\right)^2}$ | $\dfrac{Gm_1m_2}{(-x_1+x_2)^2+(-y_1+y_2)^2+(-z_1+z_2)^2}$ | $\dfrac{m_0}{\sqrt{1-\frac{v^2}{c^2}}}$ |
| $x_1y_1 + x_2y_2 + x_3y_3$ | $Nn\mu$ | $\dfrac{q_1q_2}{4\pi\epsilon r^2}$ |
| $\dfrac{q_1}{4\pi\epsilon r^2}$ | $Efq_2$ | $q\left(Bv\sin\left(\theta\right) + Ef\right)$ |
| $\dfrac{m\left(u^2+v^2+w^2\right)}{2}$ | $Gm_1m_2\left(\frac{1}{r_2} - \frac{1}{r_1}\right)$ | $gmz$ |
| $\dfrac{k_{spring}x^2}{2}$ | $\dfrac{-tu+x}{\sqrt{1-\frac{u^2}{c^2}}}$ | $\dfrac{t-\frac{ux}{c^2}}{\sqrt{1-\frac{u^2}{c^2}}}$ |
| $\dfrac{m_0v}{\sqrt{1-\frac{v^2}{c^2}}}$ | $\dfrac{u+v}{1+\frac{uv}{c^2}}$ | $\dfrac{m_1r_1+m_2r_2}{m_1+m_2}$ |
| $Fr\sin\left(\theta\right)$ | $mrv\sin\left(\theta\right)$ | $\dfrac{mx^2\left(\omega^2+\omega_0^2\right)}{4}$ |
| $\dfrac{q}{C}$ | $\arcsin\left(n\sin\left(\theta_2\right)\right)$ | $\dfrac{1}{\frac{n}{d_2}+\frac{1}{d_1}}$ |
| $\dfrac{\omega}{c}$ | $\sqrt{x_1^2 - 2x_1x_2\cos\left(\theta_1-\theta_2\right) + x_2^2}$ | $\dfrac{Int_0\sin^2\left(\frac{n\theta}{2}\right)}{\sin^2\left(\frac{\theta}{2}\right)}$ |
| $\arcsin\left(\frac{lambd}{dn}\right)$ | $\dfrac{a^2q^2}{6\pi c^3\epsilon}$ | $\dfrac{4\pi Ef^2c\epsilon\omega^4r^2}{3\left(\omega^2-\omega_0^2\right)^2}$ |
| $\dfrac{Bqv}{p}$ | $\dfrac{\omega_0}{1-\frac{v}{c}}$ | $\dfrac{\omega_0\left(1+\frac{v}{c}\right)}{\sqrt{1-\frac{v^2}{c^2}}}$ |
| $\dfrac{h\omega}{2\pi}$ | $I_1 + I_2 + 2\sqrt{I_1I_2}\cos\left(\delta\right)$ | $\dfrac{\epsilon h^2}{\pi mq^2}$ |

Table 5: A few examples of equations from the Feynman dataset.

## C Does memorization occur?

It is natural to ask the following question: due to the large amount of data seen during training, is our model simply memorizing the training set ? Answering this question involves computing the number of possible functions which can be generated. To estimate this number, calculating the number of possible skeleton $N_s$ is insufficient, since a given skeleton can give rise to very different functions according to the sampling of the constants, and even for a given choice of the constants, the input points $\{x\}$ can be sampled in many different ways.

Nonetheless, we provide the lower bound $N_s$ as a function of the number of nodes in Fig. 8, using the equations provided in [24]. For small expressions (up to four operators), the number of possible expressions is lower or similar to than the number of expressions encountered during training, hence one cannot exclude the possibility that some expressions were seen several times during training, but with different realizations due to the initial conditions. However, for larger expressions, the number of possibilities is much larger, and one can safely assume that the expressions encountered at test time have not been seen during training.

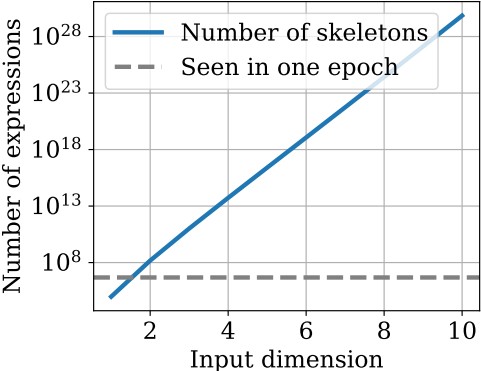

Figure 8: **Our models only see a small fraction of the possible expressions during training.** We report the number of possible skeletons for each number of operators. Even after a hundred epochs, our models have only seen a fraction of the possible expressions with more than 4 operators.

## D Attention maps

A natural question is whether self-attention based architectures are optimally suited for symbolic regression tasks. In Fig. 9, we show the attention maps produced by the encoder of our Transformer model, which contains 4 layers avec 16 attention heads (we only keep the first 8 for the sake of space). In order to make the maps readable, we consider one-dimensional inputs and sort them in ascending order.

The attention plots demonstrate the complementarity of the attention heads. Some focus on specific regions of the input, whereas others are more spread out. Some are concentrated along the diagonal (focusing on neighboring points), whereas others are concentrated along the anti-diagonal (focusing on far-away points.

Most strikingly, the particular features of the functions studied clearly stand out in the attention plots. Focus, for example, on the 7th head of layer 2. For the exponential function, it focuses on the extreme points (near -1 and 1); for the inverse function, it focuses on the singularity around the origin; for the sine function, it reflects the periodicity, with evenly spaces vertical lines. The same phenomenology can be acrossed is several other heads.

## E Additional in-domain results

Fig. 10, we present a similar ablation as Fig. 4 of the main text but using the $R^2$ score as metric rather than accuracy.

# F   Additional out-of-domain results

**Complexity-accuracy**   In Fig. 11, we display a Pareto plot comparing accuracy and formula complexity on SRBench datasets.

**Jin benchmark**   In Fig. 12, we show the predictions of our model on the functions provided in [37]. Our model gets all of them correct except for one.

**Black-box datasets**   In Fig. 13, we display the results of our model on the black-box problems of SRBench.

**Strogatz datasets**   Each of the 14 datasets from the ODE-Strogatz benchmark is the trajectory of a 2-state system following a first-order ordinary differential equation (ODE). Therefore, the input data has a very particular, time-ordered distribution, which differs significantly from that seen at train time. Unsurprisingly, Fig. 14 shows that our model performs somewhat less well to this kind of data in comparison with GP-based methods.

**Ablation on input dimension**   In Fig. 15, we show how the performance of our model depends on the dimensionality of the inputs on Feynamn and black-box datasets.

**Ablation on decoding strategy**   In Fig. 16, we display the difference in performance using two decoding strategies.

**Ablation on the use of i) mixture of distributions during training, ii) scaling during inference**
It is generally observed that Transformers struggle to generalize out-of-distribution, especially in mathematical tasks [39]. We demonstrate that both i) and ii) are necessary to handle datasets involving input distributions that are (i) neither gaussian nor uniform, and (ii) vary across wide ranges of scales.

For (i), we provide in Fig. 17 a qualitative example on a model trained with Gaussian and uniform distributions that a distribution-shift at test time can cause failure. Consider the function $y = x_1 \cos(x_0 + x_1)$. Recall the model was trained on datapoints sampled from distributions either $N(0, 1)$ or $\mathcal{U}([-1, 1])$. As we sample 100 datapoints from $\mathcal{U}([0, 6])$, we see the E2E model makes good predictions, whereas, adding 100 datapoints, sampled uniformly between $\mathcal{U}([-7, 5])$, degrades the model prediction.

For (ii), we also provide in Fig. 18 a qualitative example of failure on the same function $y = x_1 \cos(x_0 + x_1)$ and datapoints when scaling is not used. We additionally report results on SRBench evaluation for our E2E model without scaling in Table 6 and show that changes in scales during inference put transformers outside their comfort zone.

Table 6: The importance of scaling at inference for transformer-based approaches.

| Refinement | Scaler | Feynman [mean R2>0.99] | Black-box [median R2] |
|:---:|:---:|:---:|:---:|
| With | With | **0.84** | **0.87** |
| Without | With | 0.78 | 0.70 |
| With | Without | 0.53 | 0.64 |
| Without | Without | 0.06 | 0.46 |

Results on SRBench show that scaling is necessary to achieve competitive results. Note that refinement of constants can improve the performance of the unscaled prediction, however it is not enough to even catch up with the E2E without refinement model.

# G   Extended comparison with prior work.

SymbolicGPT [5] and NSRS [6] both train Transformers to predict function skeletons with other tokenization strategies. SymbolicGPT is prone to training instabilities when considering functions with high value variations and NSRS' architecture is not able to scale to high dimensions because the tokenized input grows linearly with the input dimension. [29] also predicts skeletons but focus on

the problem of inferring one-dimensional recurrence relations from small sets of points in case only, while we estimate functions of many variables over larger sets of points.

Only NSRS [6] provides a pre-trained model, but it was only trained on problems with dimensions $\leq 3$, corresponding to a very small subset of SRBench. Note however that even at these low dimensions, NSRS seems to perform less well than our model: the authors report an accuracy (defined at R2$> 0.95$) on the Feynman datasets of $\approx 0.75$ in their appendix (Fig. 9), whereas we get $\approx 0.84$ on R2$> 0.99$ on all dimensions.

The benchmark we used for our comparison, SRBench, is currently the most extensive and up-to-date benchmark for SR, and provides comparisons with other DL-based methods such as DSR [20]. Note also that the ablation of Tables 1 and 2 and Fig. 4 (in-domain), as well as Figures 5 and 13 (out-of-domain) are provided to show the benefit of the E2E approach over skeleton approaches.

Note that an other end-to-end approach[40], very similar to ours, was released two months after us.

## H   Extension of our model to dimension $> 10$.

Our method still remains improvable in scaling to larger dimensions. The reason we restricted our model to dimension $\leq 10$ is that the input sequence length becomes prohibitively long beyond, and that generating high-dimensional functions in an unbiased way becomes increasingly tricky. Nonetheless, since the objective of SR is to output interpretable formulas, we argue that SR is most useful for moderately low dimensional problems. For example, $1 - 10$ dimensional problems already cover a large class of physical systems : for instance, point objects can be represented by their position, speed and mass, 7 parameters. Additionally, in many real world problems where more than 10 features are available, some of the features are often irrelevant or heavily correlated. To mitigate this, one typically carries out feature selection before modeling the data.

We tested our model on the high-dimensional problems of SRBench (up to 1000 input dimensions), by feeding to our model only the 10 features most correlated with the output. This naive strategy already obtained encouraging results (with a median R2 score of $0.72$, to compare with $0.58$ for DSR and $0.55$ for gplearn, but still well below Operon which stands at $0.91$).

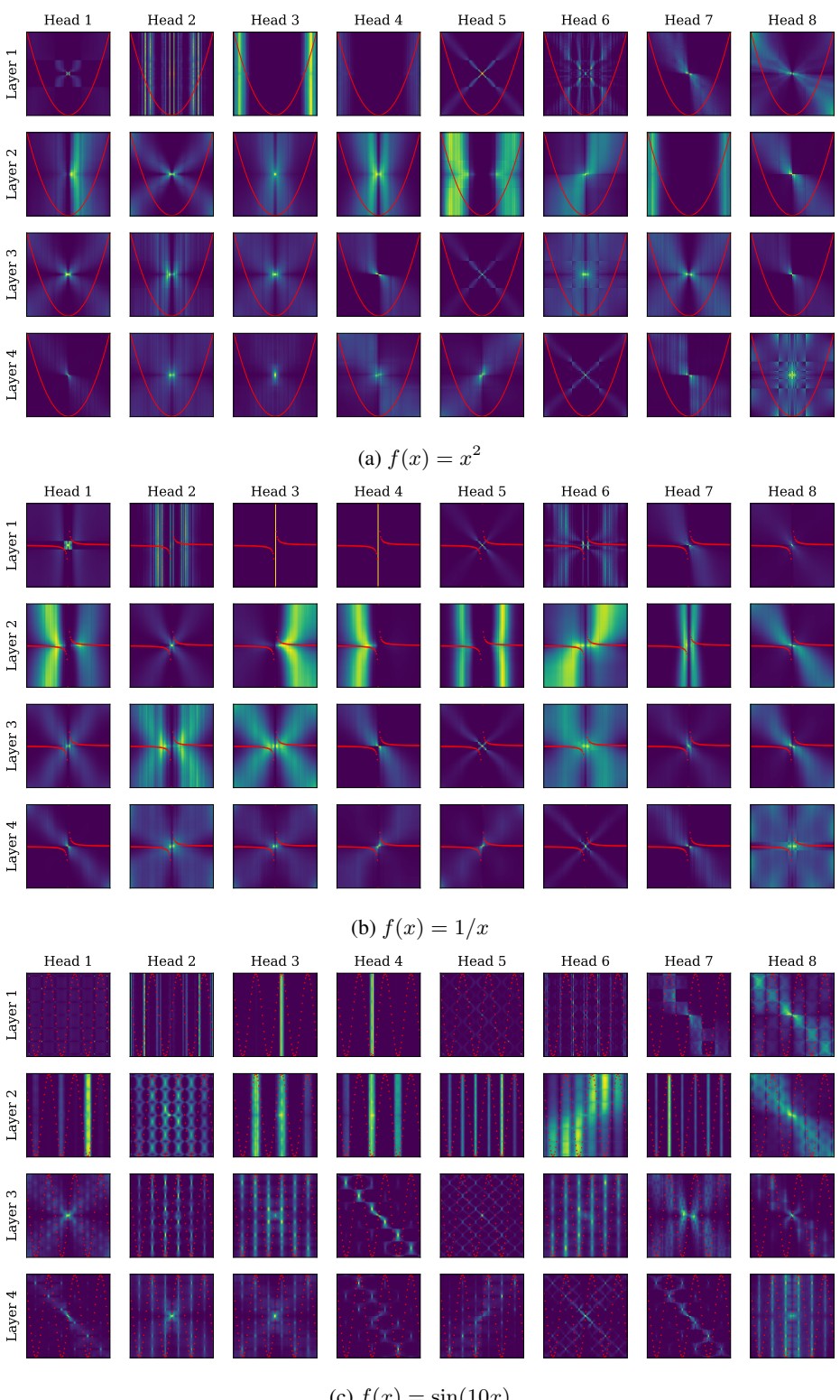

(a) $f(x) = x^2$

(b) $f(x) = 1/x$

(c) $f(x) = \sin(10x)$

Figure 9: **Attention maps reveal distinctive features of the functions considered.** We presented the model 1-dimensional functions with 100 input points sorted in ascending order, in order to better visualize the attention. We plotted the self-attention maps of the first 8 (out of 16) heads of the Transformer encoder, across all four layers. We see very distinctive patterns appears: exploding areas for the exponential, the singularity at zero for the inverse function, and the periodicity of the sine function.

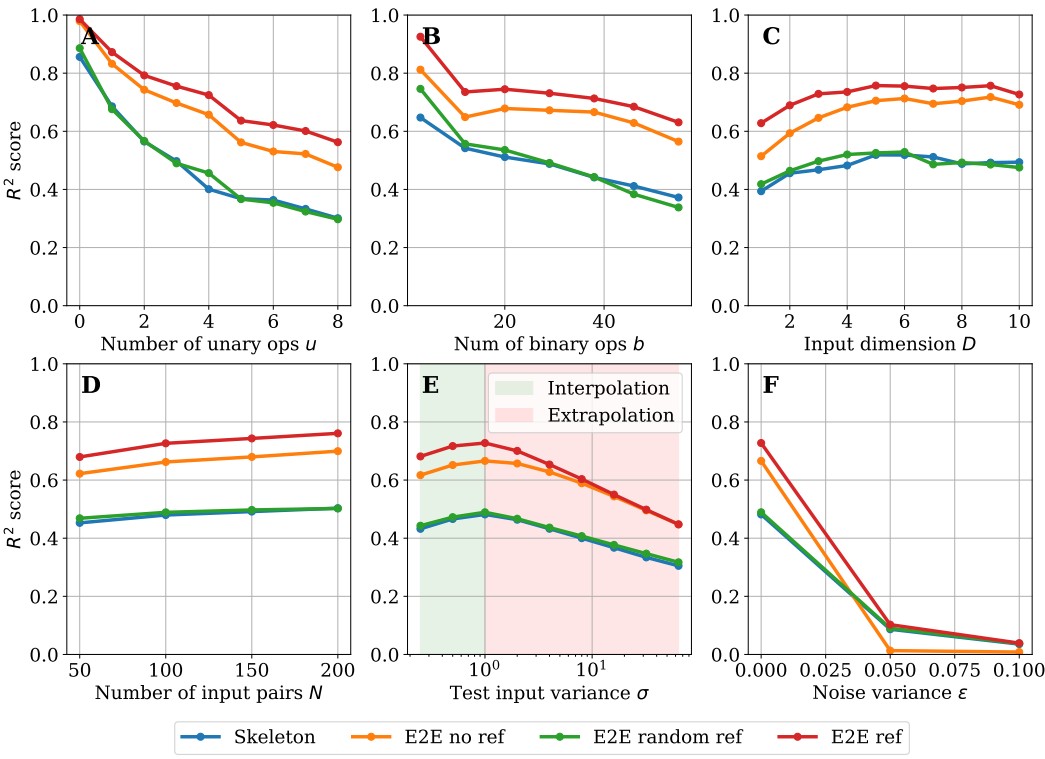

Figure 10: **Ablation over the function difficulty (top row) and input difficulty (bottom row).** We plot the $R^2$ score (Eq. 1). **A:** number of unary operators. **B:** number of binary operators. **C:** input dimension. **D:** Low-resource performance, evaluated by varying the number of input points. **E:** Extrapolation performance, evaluated by varying the variance of the inputs. **F:** Robustness to noise, evaluated by varying the multiplicative noise added to the labels.

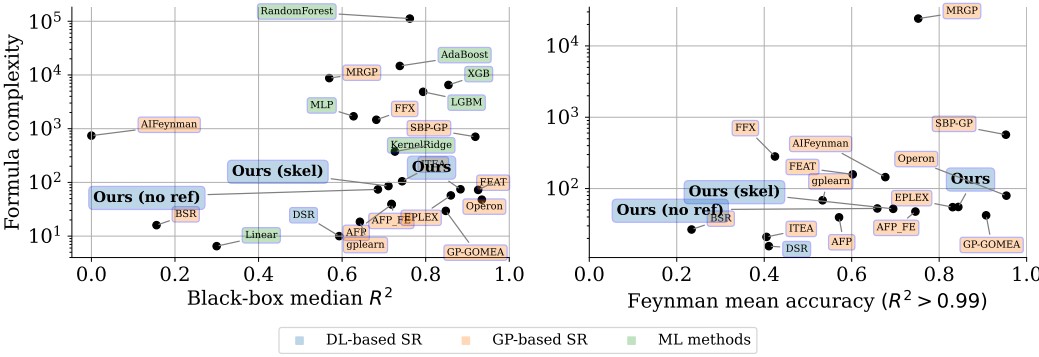

Figure 11: **Complexity-accuracy pareto plot.** Pareto plot comparing the average test performance and formula complexity of our models with baselines provided by the SRbench benchmark [7], both on Feynman SR problems [1] and black-box regression problems. We use colors to distinguish three families of models: deep-learning based SR, genetic programming-based SR and classic machine learning methods (which do not provide an interpretable solution).

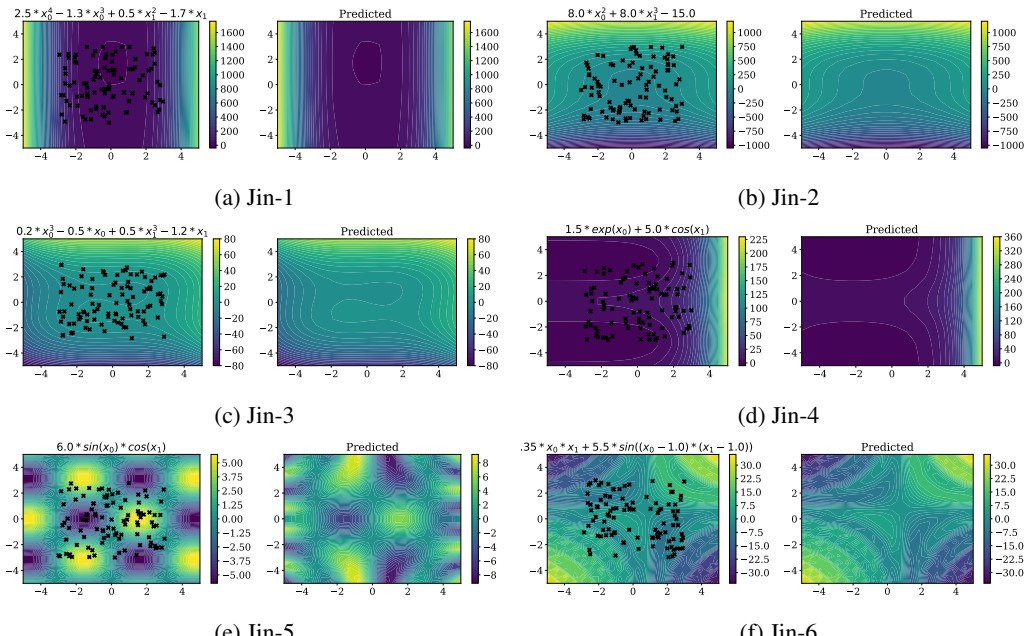

(a) Jin-1        (b) Jin-2

(c) Jin-3        (d) Jin-4

(e) Jin-5        (f) Jin-6

Figure 12: **Illustration of our model on a few benchmark datasets from the litterature.** We show the prediction of our model on six 2-dimensional datasets presented in [37] and used as a comparison point in a few recent works [38]. The input points are marked as black crosses. Our model retrieves the correct expression in all but one of the cases: in Jin5, the prediction matches the input points correctly, but extrapolates badly.

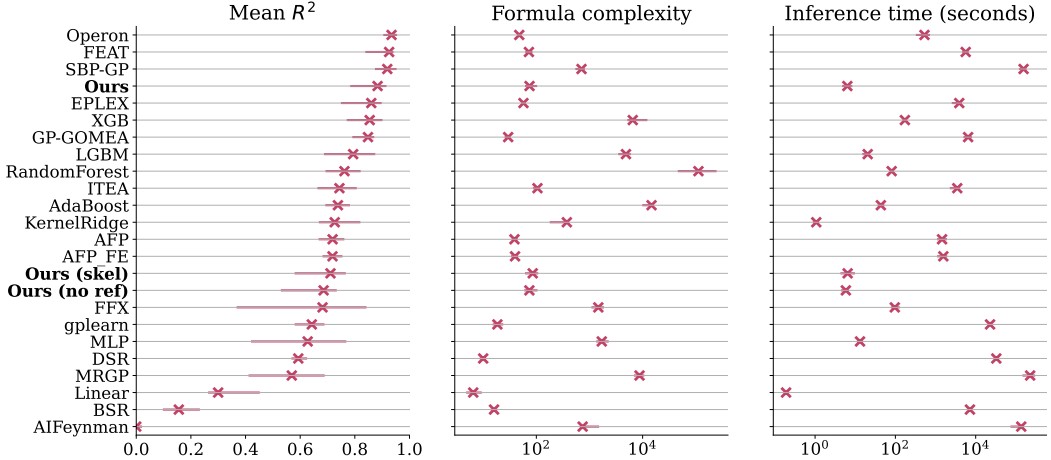

Figure 13: **Performance metrics on black-box datasets.**

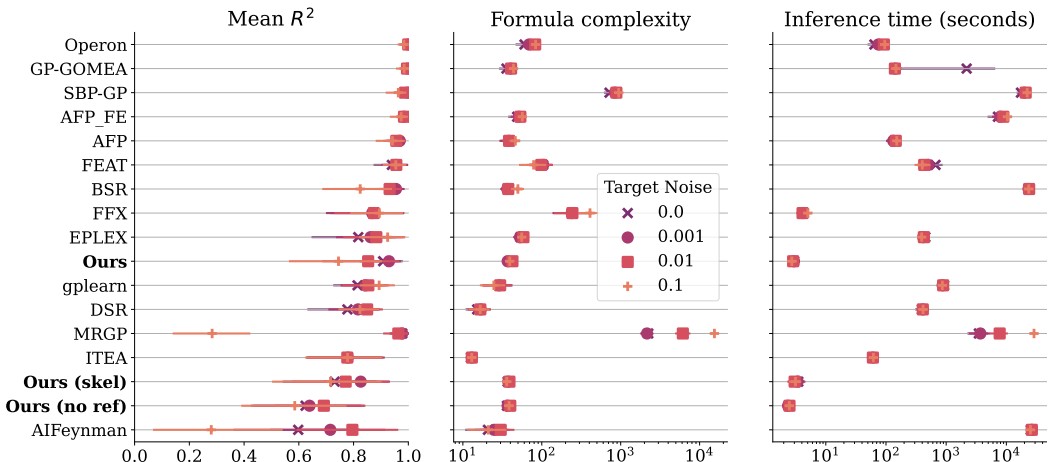

Figure 14: **Performance metrics on Strogatz datasets.**

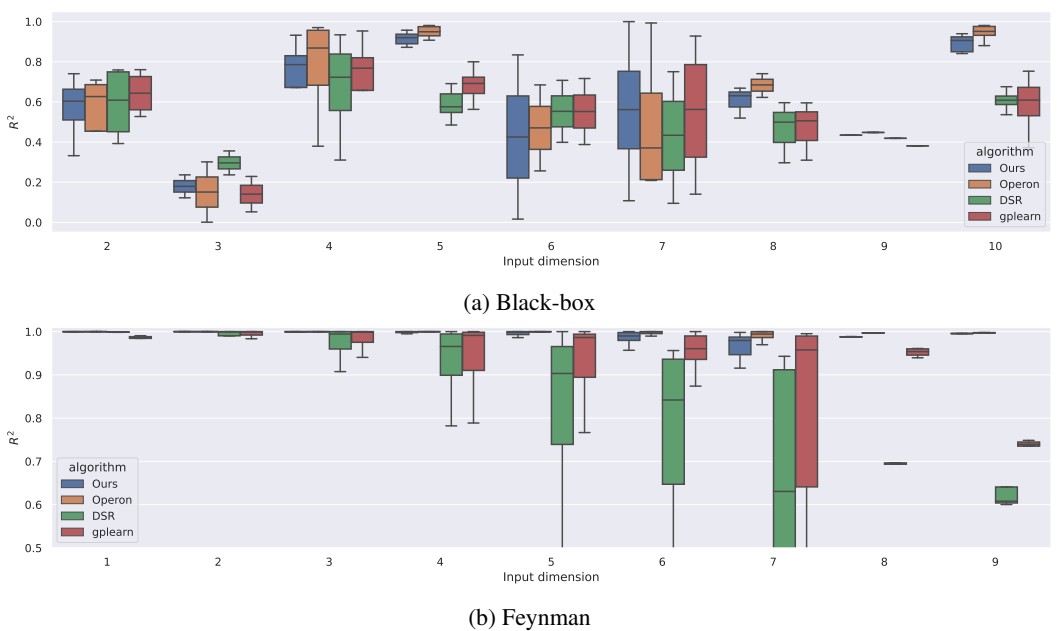

(a) Black-box

(b) Feynman

Figure 15: **Performance metrics on SRBench, separated by input dimension.**

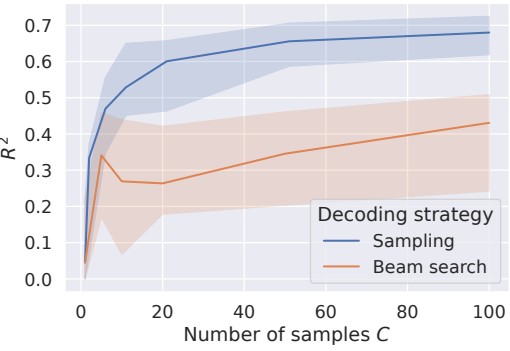

Figure 16: **Median $R^2$ of our method without refinement on black-box datasets when $B = 1$, varying the number of decoded function samples.** The beam search [34] used in [6] leads to low-diversity candidates in our setup due to expressions differing only by small modifications of the coefficients.

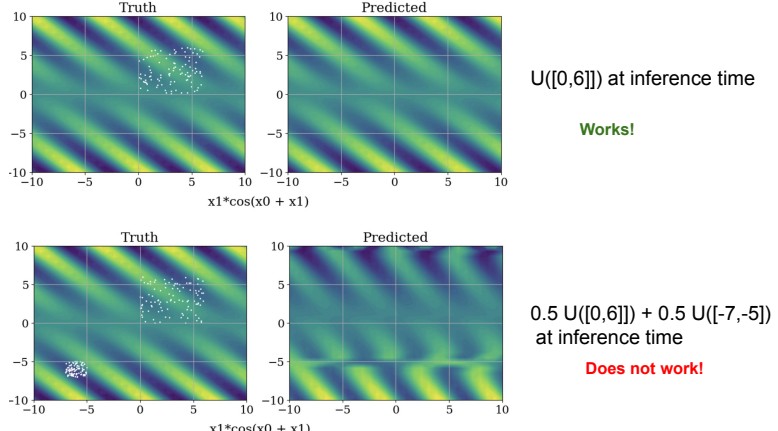

Figure 17: **Transformers do not generalize well to distribution-shift.**

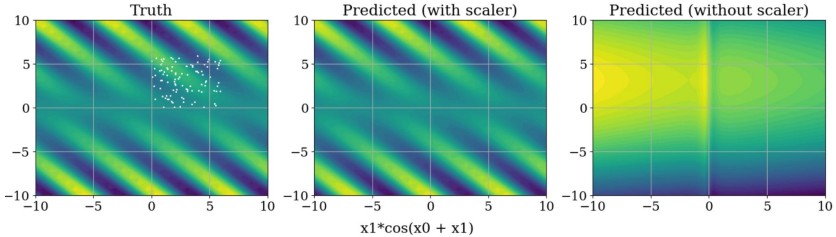

Figure 18: **Transformers do not generalize well to scale-shift.**