# OpenReview forum: "End-to-end Symbolic Regression with Transformers"
_NeurIPS.cc/2022/Conference — NeurIPS 2022 Accept_

### Official Review · Reviewer_ye2k · 2022-07-09

**Rating:** 7
**Confidence:** 5
**Soundness:** 4 excellent
**Presentation:** 3 good
**Contribution:** 3 good

**Summary:**

This paper provide a novel sequence prediction based solution to symbolic regression (SR). Compared to search-based SR methods, the proposed method is more efficient in inference and generalize to new tasks effortlessly. Compared to privious sequence based SR methods, the proposed E2E is able to scale to larger dimensions.

**Questions:**

One important issue for the sequence based SR is that people need to know to what extent the model is learned to overfit the pretraining data. The authors provided OOD evaluations in the exp section  and the dataset generation controlling params in the supples, which is helpful to this question. However, it would be better to 1) expliclitely show how different is the OOD symbolic samples differ from the pre-training data generation params? Better demonstrate this with concrete examples of OOD symbolic ground truth and  pretraining examples. And 2) it is helpful but not mandatory to: show that modifying the pre-training dataset distribution could positively contribute to but not greatly impact the OOD result.


As a new method targeting at sequence based SR, it is better to compare with prior works in this domain, which are symbolicGPT and NSRS. I am aware that official implementations seems to be not yet available, hence I'm not posting this as a must-have. However, since the E2E SR and symbolicGPT are similar in many ways, it would still be beneficial if we could see the simulated results: one way is to alter the E2E implementation so as to only predict skeletons, then apply coeff optimization to un-officially simulate the symbolicGPT.

**Limitations:**

This work does not explicitely show potential negative societal impact.


**Strengths And Weaknesses:**


Symbolic regression is one challenging, theoretically and practically important task that incorperates pattern recognition, rule based reasoning, sequence modeling, discrete optimization, and beyond.

Compared to GP based SR, the sequence prediction based SR scales to novel unseen problems effortlessly. Compared to previous works in the sequence prediction based SR such as symbolicGPT and NSRS, this work scales better, capable to deal with > 3 dimensional problems and is with better accuracy. This paper claims that using the proposed dataset generation scheme, the pre-trained E2E SR could generalize to both in-distribution and OOD problems.

In the formulation of sequence based SR, the most critical issue is how to sample the pre-training tasks such that the model learned on the generated samples could capture symbolic knowledge, while not overfit to the pre-training dataset. To address this, the authors provided a principled dataset generation workflow, and showed via experiments that E2E could generalize to OOD. The author also provided in the supples additional series of experiments on whether the E2E is memorize the data.


The authors provided implementations which helps the reproducibility.

Though being a new SR paper that pushes the SOTA, this paper upscales the dimension from (<=3) only to (<10), as the author mentioned in line 287. Hence the method still remains improvable in scaling to larger dimensions.

Other limitations dwell in the evaluation of the pre-training data generation procedure, which is critical to this sequence prediction type of SR formulation, but is under evaluated.

I haven't checked the provided implementations carefully. If the results are fully reproducible and more evaluations could be given to pre-training data generation on OOD, based on my limited assessment, this paper could be given high merit.

---

> ### Author Response · Authors · 2022-08-02
> **Rebuttal**
>
> We thank the reviewer for their valuable comments and strong endorsement of our paper. Please find our response below.
>
> ***Though being a new SR paper that pushes the SOTA, this paper upscales the dimension from (<=3) only to (<10), as the author mentioned in line 287. Hence the method still remains improvable in scaling to larger dimensions***
> We agree with this assessment, and will mention this limitation more explicitly. The reason we restricted our model to dimension <=10 is that the input sequence length becomes prohibitively long beyond, and that generating high-dimensional functions in an unbiased way becomes increasingly tricky.
> Nonetheless, since the objective of SR is to output interpretable formulas, we argue that SR is most useful for moderately low dimensional problems. For example, 1- 10 dimensional problems already cover a large class of physical systems : for instance, point objects can be represented by their position, speed and mass, 7 parameters.
>
> Additionally, in many real world problems where more than 10 features are available, some of the features are often irrelevant or heavily correlated. To mitigate this, one typically carries out feature selection before modeling the data. Motivated by this comment, we tested our model on the high-dimensional problems of SRBench, by feeding to our model only the the 10 features most correlated with the output. This naive strategy already obtained encouraging results (with a median R2 score of 0.72, to compare with 0.58 for DSR and 0.55 for gplearn, but still well below Operon which stands at 0.91​​), which we shall include in the main paper.
>
> ***Other limitations dwell in the evaluation of the pre-training data generation procedure, which is critical to this sequence prediction type of SR formulation, but is under-evaluated***
> For the dimensions considered here, the data generation procedure is rather standard, and has been used in most previous work. Our out-of-domain results on SRBench, suggest that it is sufficient to achieve good performance on black-box datasets.
> However, we agree that a better understanding of the role of data generation procedure is an important next step, if we want pre-training based approaches to SR to become better – we will add a discussion on this in the future work section.
>
> ***It would be better to explicitly show how different is the OOD symbolic samples differ from the pre-training data generation params? Better demonstrate this with concrete examples of OOD symbolic ground truth and pre-training examples.***
> This is a very good point, and we acknowledge that concrete examples would help a lot the reader. Hence, we added a section in the Appendix (section F), where we provide two tables comparing typical expressions from our random generator with some from the Feynman dataset. From this, it is clear that our expressions tend to be more complex ; note that to remain unbiased towards the OOD datasets, we did not explicitly design our generator to “look like” the Feynman expressions.
>
> We would like to emphasize the fact that the generalization of our model cannot be attributed to mere memorization. Indeed, (i) the number of possible skeletons is much larger than the number seen during training as shown in App. C ; (ii) even when a skeleton has already been seen, the function surely differs by the value of the constants, which affect the shape of the function significantly ; (iii) even for two identical functions, the points at which they are observed are never the same.
>
> ***It is helpful but not mandatory to show that modifying the pre-training dataset distribution could positively contribute to but not greatly impact the OOD result***
> We agree that a better understanding of the role of data generation procedure is an important next step, if we want pre-training based approaches to SR to become better – we will add a discussion on this in the future work section.
>
> ***It is better to compare with prior works in this domain, which are symbolicGPT and NSRS. I am aware that official implementations seems to be not yet available, hence I'm not posting this as a must-have. One way is to alter the E2E implementation so as to only predict skeletons, then apply coeff optimization to un-officially simulate the symbolicGPT.***
> We actually do provide results of the skeleton-only counterpart to our model in tables 1 and 2 (for in-domain) and Fig. 1 and 5 (for out-of-domain, see "Ours (skel)”), which perform significantly less well that the E2E model.
>
> Note that symbolicGPT and NSRS are currently limited to very low-dimensional problems, so it is impossible to evaluate them on SRBench. However, even in dimension <=3, NSRS seems to perform less well than our model : the authors report an accuracy (R2>0.95) of ~0.75 on the feynman datasets in their appendix (Fig. 9), whereas we get ~0.84 (R2>0.99) on all dimensions.

---

### Official Review · Reviewer_LNce · 2022-07-10

**Rating:** 5
**Confidence:** 3
**Soundness:** 2 fair
**Presentation:** 3 good
**Contribution:** 2 fair

**Summary:**

This paper proposed a transformer-based approach to perform end-to-end symbolic regression, that is, predicting simultaneously the function and the values of numerical constants. The trained model is augmented with several tricks at inference, including refinement, scaling, and bagging, etc. Experimental results show that the accuracy approaches to the state-of-the-art GP approach and the inference time is significantly reduced by several orders of magnitude.

**Questions:**

1. Could you make a further comparison with [29] instead of just mentioning “[29] infers one-dimensional recurrence relations”? It seems that both of them have the same architecture and similar tasks.

2. Could you please provide more ablation studies to demonstrate the effects of: (1) mixture of distributions and different distribution shapes; (2) input points scaling at inference time.

3. Could you please compare with other deep learning symbolic methods like [5] in experiments? It seems that [5] performs well on both running time and accuracy.

**Limitations:**

Please see questions listed above.

**Strengths And Weaknesses:**

Symbolic regression is well studied with various machine learning technique. In contrast to addressing the problem in a two-step manner, the proposed approach directly predicts the function along with the value of constants. Such an end-to-end model is not surprisingly new. The key innovation is the employed tricks for improving the performance of trained model, and the employed techniques are technically sound.

As the accuracy is still lower than state-of-the-art GP approaches, the main benefit of the proposed approach is the reduced inference time. However, I am not sure whether inference time is the key factor of symbolic regression. It would be helpful to introduce more background on the importance of inference time.

Besides, more comparison with other end-to-end symbolic regression approach and ablation studies would be quite helpful to demonstrate the effectiveness of the proposed approach.

---

> ### Author Response · Authors · 2022-08-02
> **Rebuttal [1/2]**
>
> We thank the reviewer for their constructive comments and suggestions. Due to space limitations, we split our answers in two different comments.
>
> ***On the importance of inference time***
> We would like to thank the reviewer for pointing out the need for a discussion on inference time. This property has been neglected by the SR community, because the inference time for most GP algorithms is both long and unbounded (the longer, the better the results). Yet we are convinced that it is important for two reasons.
>
> First, recall that the search space in SR is very large. A first pass through a fast symbolic regressor could reduce the search space, by performing feature or operator selection, or serve as an initialisation step to a GP approach (by populating its initial guesses). Such an approach has been proposed in DSR [1].
>
> Second, SR is still a research topic with just a few practical applications, but we believe it will eventually be used on a wider range of tasks.  It is already a viable alternative (with better interpretability) to current algorithms on regression tasks, as SRBench showed some SR techniques outperformed classical ones e.g. decision trees, XGBoost or NN on black-box datasets. The importance of inference speed will grow as SR is applied to real-time tasks. For instance, in control or reinforcement learning (RL), an agent might want to estimate (via SR) a model of its environment to optimize future actions (see [2,3]). We will include this discussion in the main paper.
>
> [1] Petersen et al. "Deep symbolic regression: Recovering mathematical expressions from data via risk-seeking policy gradients.", 2019.
>
> [2] Kubalík et al. "Symbolic regression methods for reinforcement learning", 2021.
>
> [3] Derner et al. "Symbolic regression for constructing analytic models in reinforcement learning", 2019.
>
> ***Comparison with [d’Ascoli et al., 2021]***
> Although the general pre-training method is similar, there are a number of differences, which we will better develop in the revised version.
>
> First, the tasks are very different. [D’Ascoli et al., 2021] focuses on the one-dimensional case only, but they study a harder problem : inferring recurrence relations from small sets of points, while we estimate functions of many variables over larger sets of points.
>
> Second, we introduce novel methods to handle high-dimensional data - an embedding module to compress the inputs, the mixed numeric-symbolic vocabulary for constant prediction, as well as various other tricks.
>
>
> ***The accuracy is still lower than state-of-the-art GP approaches***
> We believe that our model, while not strictly SOTA in terms of raw accuracy, is the first DL-based approach to strike an excellent balance between the desirable properties of a SR algorithm (accuracy, simplicity, inference time). As Fig. 5 shows, we are close to the best GP algorithms in terms of overall accuracy. On the Feynman dataset, we achieve high accuracy while producing simpler functions than many GP approaches (see Fig. 11).

---

> > ### Author Response · Authors · 2022-08-02
> > **Rebuttal [2/2]**
> >
> > ***Comparison with other end-to-end approaches and with NSRS***
> > To the best of our knowledge, no other end-to-end (E2E) approach for SR existed at the time of submission – all existing methods predicted skeletons. Another end-to-end technique [4] (without our inference tricks) was released two months ago, after the submission deadline.
> > As for NSRS [5], it is only applied to problems with dimension <=3, which means one can only test it on a very small subset of SRBench. Note however that even at these low dimensions, NSRS seems to perform less well than our model : the authors report an accuracy (defined at R2>0.95) on the feynman datasets of ~0.75  in their appendix (Fig. 9), whereas we get ~0.84 on R2>0.99 on all dimensions.
> >
> > The benchmark we used for our comparison, SRBench, is currently the most extensive and up-to-date benchmark for SR, and provides comparisons with other DL-based methods such as DSR [1]. Note also that the ablation of Tables 1 & 2 (in-domain) and Figures 5 & 11 are provided to show the benefit of the E2E approach over methods such as NSRS.
> >
> > [4] Vastl, Martin, et al. "SymFormer: End-to-end symbolic regression using transformer-based architecture.", 2022.
> >
> > [5] Biggio et al. "Neural symbolic regression that scales.", 2021.
> >
> > ***Ablation of (1) mixture of distributions and (2) input points scaling at inference time.***
> > We thank the reviewer for bringing up this important point, which indeed needs to be expanded. It is generally observed that Transformers struggle to generalize out-of-distribution, especially in mathematical tasks [6]. Hence, (1) and (2) are necessary to handle datasets involving input distributions that are (1) neither gaussian nor uniform, and (2) vary across wide ranges of scales.
> >
> > [6] Welleck et al. "Symbolic Brittleness in Sequence Models: on Systematic Generalization in Symbolic Mathematics", 2021.
> >
> > For (1), we will add complete ablations in the camera ready copy as the corresponding experiments are too long to be run during the rebuttal period (we need to train a model without mixture from scratch).
> > However, we used a model trained on a generator with uniform input distribution for only a few epochs to provide a qualitative example that shows how distribution-shift at test time can cause failure (see figure “rebuttal/mixture_ablation.pdf” in the SM).
> >
> > Consider the function cos(x0+x1)*x1. Recall the model was trained on distributions either N(0,1) or U([-1,1]). As we sample 100 datapoints from U([0,6]), we see the E2E model makes good predictions, whereas, adding 100 datapoints, sampled uniformly between  U([-7,5]), degrades the model prediction.
> >
> > For (2), since the rescaling happens at inference, we ran the SRBench evaluation for our E2E model without scaling, and as expected got worse results (see table below, which shall be added to the Appendix). We also added the figure “rebuttal/rescale_ablation.pdf” in the SM to provide a qualitative example of failure when the scaler is not used.
> >
> > |Refinement|Scaler|Feynman [mean R2>0.99]|Black-box [median R2]|
> > |-|-|-|-|
> > |With|With| 0.84|0.87|
> > |Without|With|0.78|0.70|
> > |With|Without|0.53|0.64|
> > |Without|Without|0.06|0.46|

---

> > > ### Comment · Reviewer_LNce · 2022-08-09
> > > **Thank you for the rebuttal**
> > >
> > > Thank you for adding the comparison with other approaches and the ablation study. If the final submission does contain such changes, I would be happy to raise my score.

---

> > > > ### Author Response · Authors · 2022-08-09
> > > > **Thank you for the discussion**
> > > >
> > > > Due to time constraints with respect to today’s deadline, we included in the appendix the discussion on inference time (App. G), an extended comparison with other DL skeleton approaches (App. H), as well as the ablation studies (App. E). We commit to integrate them as well as possible to the main paper in the camera ready version, if the paper gets acceptance.
> > > > We would like to thank the reviewer for changing his score, as well as providing a set of questions that will give more impact to our work.

---

### Official Review · Reviewer_Hxn5 · 2022-07-12

**Rating:** 6
**Confidence:** 1
**Soundness:** 3 good
**Presentation:** 2 fair
**Contribution:** 3 good

**Summary:**

This paper proposes a symbolic regression model using Transformers. It enables to pre-train the model using datasets from the other domain, which is helpful when the target domain dataset is relatively small. Moreover, this approach performs the parameter refinement with BFGS to further enhance the performance. Compared to the state-of-the-art approaches, which are mainly genetic programming based, this approach achieved not superior but competitive performance on benchmark tasks, and showed significantly smaller inference time.


**Questions:**

Please see the comment above.

**Limitations:**

Although the checklist says the code is available, I couldn't find the pointer to the code in the main text. It would always be nice to have a pointer to the code, even if it is included in the supplementary.

Algorithmic description of the proposed approach would help readers understand.

**Strengths And Weaknesses:**

Clarity

The motivation of each algorithmic component is described well. However, as I am not an expert in SR, it was hard for me to understand this paper in details. Probably it is because of my lack of the common knowledge in the SR community. For example, how is it pretrained? I couldn’t find any statement about the setting for the pretraining. Or is the method written in Section 2.1 for the pretraining? If so how is the post-training performed? The computational time written in 166 is for what? It should depends on the dataset, right? It was also not clear for me how the refinement of the parameter is performed in the end-to-end manner.

Significance

Based on the authors’ statement, this approach is the first one that reaches competitive performance to GP-based approaches. Although the performance is not superior, the advantage of using pre-training over GP-based approach is promising.

The author’s claim the advantages of the proposed approach over the top performance approaches in its inference time. However, it was not evaluated whether this comes from the trick described in Section 2.2 or it is intrinsic to the approach using Transformer.

---

> ### Author Response · Authors · 2022-08-02
> **Rebuttal**
>
> We thank the reviewer for their useful comments and suggestions. Please find our answers below.
>
> ***For example, how is it pretrained? I couldn’t find any statement about the setting for the pretraining. Or is the method written in Section 2.1 for the pretraining? If so how is the post-training performed?***
> We acknowledge that the term “pre-training” can cause confusion, and we will change the wording. To clarify : our model is not pre- and then post-trained – there is only one training phase, which is performed on synthetic (i.e. randomly generated) data, where for each example the model learns to predict a function from its values at a set of points. At inference, a new set of points is presented and the trained model is tasked to predict the corresponding function.
> The synthetic training data and its generation are described in section 1. The model and its training are described in section 2.1. Finally, the inference (evaluation of unknown functions with the trained model) is described in section 2.2.
>
> ***The computational time written in line 66 is for what?***
> This is the average training time for one epoch (300k examples). Training lasts 50 epochs, at which point the accuracy on the validation set saturates.  The inference time (time needed for the trained model to predict a single function) depends on the size of the input data, but is usually around 1 second. We will clarify this.
>
> ***It was also not clear for me how the refinement of the parameter is performed in the end-to-end manner.***
> Our end-to-end denomination refers to the following fact. Previous methods predict a function skeleton, e.g. A cos(Bx+C), where A,B,C are free parameters which must then be determined by external tools. Our model provides an estimate of both the skeleton and the parameters (e.g. 2.1 cos (0.5 x +3) ), which makes it end-to-end. Parameter refinement is an optional step which improves accuracy. We will also clarify this.
>
> ***The author’s claim the advantages of the proposed approach over the top performance approaches in its inference time.  However, it was not evaluated whether this comes from the trick described in Section 2.2 or it is intrinsic to the approach using Transformer.***
> Prediction is faster with a transformer because it is trained in advance, on many synthetic examples, to predict a function from its values.
> At inference, it only needs to evaluate (a single forward pass) the learned model on the set of points it is given: a very fast process.
> Genetic algorithms, on the other hand, have no such pre-inference training phase. They predict any new function from scratch, and do not capitalize on past examples. This results in a much slower process.
> The improvements proposed in section 2.2 actually make inference slower, by resorting to external minimization techniques (BFGS) and working on ensembles, but the overall inference time remains orders of magnitude smaller than that of genetic algorithms (see figure 1).
>
> ***Although the checklist says the code is available, I couldn't find the pointer to the code in the main text. It would always be nice to have a pointer to the code, even if it is included in the supplementary.***
> Thanks for pointing this out. We will add a reference to the code in the introduction (pointing to supplementary material during the review, then to an online repository should the paper be published).

---

### Meta-Review · Area_Chair_d4FB · 2022-08-26

**Recommendation:** Accept
**Confidence:** Less certain

**Metareview:**

The paper proposes a transformer-based approach to perform end-to-end symbolic regression. All three reviewers seem to agree on the usefulness of the proposed approach to reduce inference time.  As pointed out by Reviewer Hxn5, although the performance is not superior, the advantage of using pre-training over GP-based approach is promising. The discussion phase has allowed covering important criticisms and the autors have included in the appendix a discussion on inference time (App. G), an extended comparison with other DL skeleton approaches (App. H), as well as ablation studies (App. E), commiting in turn to integrate the latter as well as possible to the main paper in the camera ready version. Furthermore it was stressed in the discussion by one of the referees that the implementation released in the supplementals is of high quality and bug-free, which should be of help to the future research community once open-sourced. For all these reasons, I am recommending the paper to be accepted.

**Award:**

No

---

### Decision · Program_Chairs · 2022-09-14

Accept